# Consistent Effects of Whey Protein Fortification on Consumer Perception and Liking of Solid Food Matrices (Cakes and Biscuits) Regardless of Age and Saliva Flow

**DOI:** 10.3390/foods9091328

**Published:** 2020-09-21

**Authors:** Victoria Norton, Stella Lignou, Stephanie P. Bull, Margot A. Gosney, Lisa Methven

**Affiliations:** 1Department of Food and Nutritional Sciences, Harry Nursten Building, University of Reading, Whiteknights, Reading RG6 6DZ, UK; v.l.norton@pgr.reading.ac.uk (V.N.); s.lignou@reading.ac.uk (S.L.); s.p.bull@reading.ac.uk (S.P.B.); 2Royal Berkshire NHS Foundation Trust, London Road, Reading RG1 5AN, UK; m.a.gosney@reading.ac.uk

**Keywords:** protein fortified foods, mouthdrying, older adults, whey protein, saliva flow

## Abstract

Although there are numerous high protein products on the market, they are typically not designed with, or for, older consumers. This is surprising considering that dietary guidelines recognise the need for higher protein intake in later life. Protein fortified products are, however, associated with negative sensory attributes and poor consumer acceptance. This paper investigates the extent of mouthdrying sensations within a high protein solid food matrix, along with the effect of age and saliva flow. Solid models using cakes and biscuits, with or without protein fortification, were investigated. The sensory profile and physical properties were analysed and two volunteer studies (*n* = 84; *n* = 70) were carried out using two age groups (18–30; 65+). Volunteers rated individual perception and liking of products, and salivary flow rates (mL/min) were measured. Unstimulated salivary flow rates were significantly lower (*p* < 0.05) in older adults, although this was not found to influence product perception. Protein fortification of cakes and biscuits significantly increased (*p* < 0.05) perceived mouthdrying, hardness and “off” flavours, and significantly reduced (*p* < 0.05) melting rate, moistness and liking compared with the control versions. There is a clear need to address negative sensory attributes associated with protein fortification of cakes and biscuits to ensure product suitability for older adults.

## 1. Introduction

Older adults are at risk of poor nutritional status due to age related changes, such as anorexia of ageing (a physiological decline in food intake with age), changes in sensory sensitivity, and oral impairments (dysphagia, tooth loss and decreased salivary flow), all of which can influence an individual’s food intake and increase the risk of malnutrition [1,2]. Immune function is also considered to decline with age and is associated with an increased risk of infection [3]. Malnutrition covers both over- and under-nutrition; however, in older adults, undernutrition predominates, being a deficiency of both macronutrients and micronutrients [4,5]. Food intake can also be reduced in older adults due to chemosensory impairments (such as loss of taste and smell) which influence food choices and consumption [6].

Protein requirements are considered to increase with age and are associated with many positive functional outcomes. Despite this, and the many high protein products available on the market, these are typically not designed with, or for, older adults. The current UK reference nutrient intake (RNI) is 0.75 g/kg/day. However, dietary guidelines recognise the need for higher protein-intake in later life (1.0–1.2 g/kg/day) with the aim of maintaining and regaining lean body mass and function, as well as aiding in recovery from illness and helping to overcome age-related changes in protein metabolism [7,8]. Accordingly, protein fortified meals and snacks (often fortified with whey protein) can provide familiar foods to older adults in order to encourage consumption and increase energy and protein intake [9,10]. Older adults consuming familiar protein enriched products were demonstrated to have increased protein intake (1.5 ± 0.6 g/kg/day) compared with a control group (1.0 ± 0.4 g/kg/day) over a 12 week period [11]. In addition, previous research has suggested that older adults have a higher liking for cakes compared with younger adults [12], therefore supporting the use of popular products (such as cakes) to potentially increase food intake within older adults.

Protein fortification can result in “off” flavours. Whey proteins are rich in sulfur amino acids and when heated they can release sulfurous and eggy aromas which influence the subsequent flavour [13,14]. When this occurs, it results in products tasting stale or “cabbage-like” and will reduce acceptability of products [15,16]. Mouthdrying sensations can also result from protein fortification, as previously demonstrated in high protein liquid model systems, a sensation shown to increase with consumers’ age and with repeated consumption of dairy beverages and oral nutritional supplements (ONS) [17,18]. Although terms such as astringency, drying and mouthdrying are commonly used interchangeably within the literature, within this paper mouthdrying specifically refers to a drying sensation in the mouthduring or after consumption of a product. Currently the exact causes of whey protein derived mouthdrying are not fully understood and are part of our ongoing investigation [19].

Individual differences in oral physiology can potentially influence food oral processing and sensory perception [20]. Accordingly, the way an individual manipulates food in their mouth, usually described as mouth behaviour, is considered to influence food choice, texture preference and satisfaction [21,22]. Food oral processing and saliva perform key roles in the breakdown of food and sensory perception [23] and older adults are considered to consume foods more slowly and have reduced salivary flow rates compared with younger adults [24,25]. Appetite is also considered to decline with age, therefore understanding its role in sensory perception is key, e.g., ONS consumption can increase thirst from drying sensations [26]. All these issues highlight the need to understand how differences in individual perception can impact sensory perception and consumption of foods in older adults.

Currently, the influence of mouthdrying on the overall perception and liking of protein fortified solid foods and the effect of saliva flow and age on this phenomenon are unclear. It is recognised that protein fortified foods can help to alleviate malnutrition. However, they need to be acceptable and palatable. Fundamental investigation of the perception of protein in foods and of individual factors influencing this perception may result in knowledge that assists in product optimisation and, subsequent health benefits from increased consumption of protein fortified foods by older adults. This paper hypothesises that (a) protein fortification of cakes and biscuits will cause mouthdrying and reduce acceptance of products and (b) individual differences (such as age, mouth behaviour, dental status, saliva flow and appetite) will influence perception and liking of products. Accordingly, hypotheses were tested as follows:(1)A pilot study was carried out to establish whether protein fortification of cakes and biscuits causes mouthdrying, thereby reducing acceptance, and whether individual differences influence perception and liking of products. The specific objectives were (a) to analyse the sensory profile and physical properties of cakes and biscuits, (b) to evaluate perception and acceptance of cakes and biscuits, with and without protein fortification and relate these to age, dental status, mouth behaviour and salivary flow rates, and (c) to use the results to optimise products for the main study.(2)The main study aimed to further investigate protein derived mouthdrying and its relationship with product acceptance. The specific objectives were (a) to analyse the sensory profile and physical properties of the optimised cakes, (b) to evaluate the perception and acceptance of cakes with and without protein fortification after consumption of a full portion, and the influence of this on rated appetite, and (c) to related these measures to individual differences including age, dental status, mouth behaviour and salivary flow rates.

## 2. Materials and Methods

### 2.1. Overview of Pilot and Main Study

The pilot study consisted of 84 healthy male and female volunteers from two age groups (42 younger adults; 18–30 years, 24.3 ± 3.6 years and 42 older adults; over 65 years, 73.6 ± 6.2 years) who completed a single blinded randomised crossover trial involving two study visits. The main study was a single blinded randomised crossover trial, with two study visits and two tasting sessions at home, involving 70 healthy male and female volunteers from two age groups (38 younger adults; 18–30 years, 25.8 ± 3.2 and 32 older adults; over 65 years, 74.6 ± 5.7 years) (volunteer overview, Table 5 in the results section). In both studies the subject size was determined by power calculations (alpha risk = 0.05 and 80% power) based on the primary outcome measures (liking and mouthdrying). In the pilot study we estimated a difference, on a 9-point hedonic scale, of 0.8, assuming a standard deviation of 1.2. In the main study we based the calculations on the pilot study intensity ratings (0–100) and we predicted a larger difference of 12, assuming a standard deviation of 16.5. This concluded, a minimum sample size of 35 and 31, respectively were required to demonstrate a significant difference between samples. However, we doubled the sample size in order to compare both between and within age groups, as there was no additional data on which to base the power calculations. Volunteers were recruited from the local Reading, UK, area. The studies were conducted in accordance with the Declaration of Helsinki and all volunteers had the study fully explained, provided informed written consent, and were informed that all data would be anonymous, fully confidential, and of their right to withdraw. The studies received a favourable opinion for from the University of Reading Research Ethics Committee and were registered on the clinical trials database (www.clinicaltrials.gov) (pilot study: UREC 18/46 and NCT03798730 and main study: UREC 19/67 and NCT04302779).

All volunteers were screened for suitability (minimal medication, non-smoker, no food allergies or intolerances, non-diabetic and not having had either cancer, oral surgery or a stroke). Volunteers who met the inclusion criteria and were willing to take part were invited to attend study visits held at the Sensory Science Centre, University of Reading; the study overview is summarised in Figure 1. In order to control extraneous variables, volunteers refrained on the day of each study visit from tea and coffee and drank a glass of water one hour before the visit. Each individual volunteer completed all their study visits at the same time of day in a temperature-controlled room (22 °C) under artificial daylight.

### 2.2. Materials

Baking ingredients were obtained from Sainsbury’s (Reading, UK). Sil Cream 64, a 100% vegetable oil, was supplied by Silbury (Silbury, Banbury, UK) and parafilm^®^ was supplied by Sigma-Aldrich (Dorset, UK). Whey powders were provided by Volac (Volac International Ltd., Royston, UK). These consisted of whey protein concentrate (WPC, Volactive Ultra-Whey 80 Instant, providing a minimum protein content of 80%, the remaining 20% being lactose, fat, moisture and ash), whey protein isolate (WPI, Volactive Ultra-Whey 90 Instant, providing a minimum protein content of 90%, the remaining 10% being lactose, fat, moisture and ash) and whey permeate (WPe, Volactose Taw Whey Permeate, providing a minimum lactose content of 89% the remaining 11% being ash, moisture, protein and fat).

### 2.3. Solid Model Preparation

The rationale for the solid model was that high energy and protein snacks can help to alleviate malnutrition, and cakes and biscuits are suitable familiar and tasty products for this purpose. The pilot study tested cake and biscuit products, each with a control, and a protein fortified version using WPI. In the main study, the control and protein cakes were optimised in a number of ways and for a number of purposes: (a) to provide a better match of ingredients between the two versions, the control incorporated whey permeate as a minimal protein alternative in place of the whey protein powder; (b) to match the final moisture content of the cakes within 1%; (c) to replace WPI with WPC because, although WPC is lower in protein, its minor constituents (e.g., lactose and fat) may contribute positively to sensory attributes [27]; (d) to add lemon zest to potentially improve both flavour and acceptability; and (e) the cakes were formed as individual cupcakes to ensure uniformity and suitability for consumption at home. All product formulations are outlined in the Appendix A.

The recipes had been developed previously by Tsikritzi et al. [28] and the University of Reading Food Research Group and were adapted for this study. In summary, for the cake dough (pilot study), the butter and sugar were creamed until smooth, the remaining ingredients were added and mixed and batter (450 g) was weighed out into 600 mL loaf tins and baked until golden brown (30 min at 170 °C). For the biscuit dough (pilot study), the fat and sugar were creamed until smooth, the remaining ingredients were added and mixed, dough was rolled out and sheeted (thickness: 1 cm; diameter cutter: 4.5 cm) and subsequently baked (9 min at 190 °C). During the main study, cupcakes were mixed using an all-in-one method and the batter (38.2 g) was weighed out into individual paper cases (80 mm × 62.5 mm) and baked until golden brown (20 min at 170 °C, final weight 35 g). Nutritional composition was analysed (Nutritics v5.096, Dublin, Ireland) taking account of heat loss (Table 1). All samples were packaged in heat-sealed pouches (polypropylene for cakes, aluminum for biscuits), frozen at −18 °C and defrosted at room temperature before consumption. A sample (150 g) from each batch was sent for microbiological testing at an accredited laboratory (SYNLAB, Northumberland, UK).

### 2.4. Sensory Profile and Physical Properties of Cakes and Biscuits

Sensory profiling was carried out using quantitative descriptive analysis (QDA™) [29] to determine the sensory differences between the control and protein products. A screened and trained sensory panel (*n* = 12; 11 female and 1 male) was used, each member with a minimum of one years’ experience and with expertise in profiling techniques, having received at least four hours specific training on profiling protein and control products. All sensory evaluation was carried out in a temperature-controlled room (22 °C), in isolated booths, and under artificial daylight. Warm filtered water (~40 °C) was used as a palate cleanser between samples; this is considered more effective than cold water at removing fatty dairy residues from the mouth [17]. The trained panel were provided with the same products as the study volunteers (45 g cake slice, 20 g biscuit and 35 g cupcake; preparation as Section 2.3). The trained panel (cakes *n* = 12; biscuits *n* = 9; cupcakes *n* = 11) developed a consensus vocabulary identifying between 33 to 36 attributes per product across the different modalities (appearance, aroma, flavour, mouthfeel and aftertaste following a 1 min delay) as outlined in Tables 2 and 3 in the results section. All panellists scored in duplicate, for each sample, in separate sessions. Samples, coded with three-digit random numbers, were provided in a monadic sequential balanced order, with sample sets randomly allocated to panellists. Visual analogue scales (VAS) (0–100) with suitable anchors were used (Compusense Cloud Software, Guelph, ON, Canada).

The moisture content (% *w*/*w*) of the cake, biscuit and cupcake were measured using a moisture analyser (Sartorius MA150, Goettingen, Germany) and water activity (a_w_) (Hydrolab C1, West Sussex, UK) was also measured. Colour measurements, *L** (light-dark); *a** (red-green); *b** (yellow-blue), were taken from the top and bottom surface of the biscuits and from the crumb and crust of the cakes and cupcakes by colorimeter (Chroma Meter CR-400, Osaka, Japan). The hue angle (arctan (*b**/*a**)) [30] and total colour difference (dE) from the control sample were also calculated [31]:(1)$$∆E^{*}={\left[{\left(∆L^{*}\right)}^{2}+{\left(∆a^{*}\right)}^{2}+{\left(∆b^{*}\right)}^{2}\right]}^{1/2}$$

Texture profile analysis (TPA) was carried out using a texture analyser (XTPlus, Stable Micro System (SMS), Godalming, UK) equipped with a load cell of 5 kg. For the cake crumb an adapted double compression test based on previous work [32] was carried out with a cylindrical probe (SMS rig code P/75) using a test speed of 5 mms/s with 5 s delay between compression tests (compression was to 25% of original height), on a 15 mm deep slice. Parameters recorded were hardness, chewiness, springiness and cohesiveness, along with adhesiveness, resilience and gumminess for cupcakes. Biscuit analysis was carried out using a three-point bend test (SMS rig code HDP/3 PB) on the texture analyser with hardness and fracturability as parameters, at a test speed of 3.0 mm/s [33]. The height of the cakes and cupcakes and the thickness and diameter of the biscuits was measured by digital calipers. All analysis was performed in triplicate on different days on all study batches consumed by the volunteers.

### 2.5. Dental Status and Mouth Behaviour Questionnaire

Volunteers completed a dental status questionnaire adapted from the World Health Organisation’s (WHO) Oral Health Questionnaire which focused on key areas including number of teeth, dentures, functional unit counting, qualitative questions, and dental cleaning methods and frequency [34]; pictures were added to improve clarity. Volunteers also completed a validated online test using the JBMB Typing Tool to categorise individual mouth behaviour preferences; this tool grouped volunteers into four types: chewers, crunchers, “smooshers” and suckers [21,22]. In both studies, all volunteers independently completed both questionnaires during their study visits.

### 2.6. Salivary Flow Rates

During the pilot study unstimulated saliva was collected at the beginning of each visit and two replicates of stimulated saliva were collected during visit one (10 to 15 min break between collections). During the main study, saliva collection (unstimulated and stimulated) was carried out at the beginning of each study visit (10 to 15 min break between collections). Saliva collection methods were adapted from previous work [25,35]. In brief, during unstimulated saliva collection, volunteers collected saliva in their mouths and spat out saliva every time they felt the urge to swallow during a 5-min time period. Stimulated saliva was collected by volunteers spitting out saliva every time they felt the urge to swallow during a 2 min time period while chewing on parafilm^®^ (5 × 5 cm). Saliva was collected in wide lid collection tubes (60 mL). Saliva weights were monitored by weighing collection tubes before and after collection. Salivary flow rates were calculated as mL/min, using the assumption that 1 g of saliva equates to 1 mL. All saliva samples were stored on ice pending analysis.

### 2.7. Individual Perception, Appetite and Liking Ratings of Products

During the pilot study, volunteers rated liking, easiness to eat and to swallow, attribute perception and appropriateness of attribute level (Just-About-Right, JAR) of cakes and biscuits individually on an iPad, either in isolated booths (younger adults) or at a table (older adults), using Compusense Cloud Software. Samples, coded with three-digit random numbers, were provided in a monadic sequential balanced order, with sample sets randomly allocated to volunteers. Volunteers received a 45 g cake slice and a 20 g biscuit, these being considered appropriate portion sizes for older adults and volunteers evaluated the same cakes at both visits to check reliability. All volunteers were trained by a short video in how to use the generalised Labelled Magnitude Scale (gLMS), a scale ranging from no sensation (0) to strongest imaginable sensation of any kind (100) [36]. Volunteers had an enforced rest period of 45 s between samples and consumed warm filtered water (rationale as outlined in Section 2.4) before completing the same series of questions on the second sample.

During the main study, appetite ratings (hunger, thirst, desire to eat, fullness, satiety and prospective food consumption) were recorded on a 0 to 100 mm visual analogue scale (VAS), with appropriate anchors [37,38] before and after consumption of each cupcake. Volunteers additionally rated liking, easiness to eat and swallow and attribute perception for each 35 g cupcake individually at home. To avoid using multiple scale types at home, VAS scales were again used. All scoring at home was done using paper booklets, and samples were coded with three-digit random numbers and provided monadically on two separate occasions in a sequential balanced order, randomly allocated to volunteers. All volunteers first received training on how to use the VAS scales via non-food related questions. An overview of individual perception and liking measures taken is outlined in Figure 2.

### 2.8. Statistical Analysis

#### 2.8.1. Sensory Profile and Physical Properties of Cakes and Biscuits

QDA data and trained panel performance was analysed using analysis of variance (ANOVA; considered the most appropriate analysis for this type of data [39,40]) in SenPAQ (version 5.01, Qi Statistics, Berkshire, UK), where the main effects (sample and assessor) were tested against the sample by assessor interaction, with sample as fixed effect and assessor as random effect.

Physical properties data were analysed in XLSTAT (version 2019.2.2, Addinsoft, Boston, MA, USA). Normally distributed data (based on normality of residuals (Shapiro-Wilk) *p* > 0.05) were analysed using *t*-tests, and data not normally distributed were treated as nonparametric and analysed using a Mann-Whitney test.

#### 2.8.2. Pilot and Main Study

Volunteers were categorised into low, medium and high groups based on average unstimulated salivary flow rates, using tertile analysis in XLSTAT; these groupings were also used for subsequent statistical analysis. In order to test associations between age and categorical data (saliva flow rate grouping, mouth behaviour, dental status and medication), a chi-square test was carried out on contingency tables using XLSTAT. Linear mixed model analysis was carried out in SAS^®^ software (Version 9.4, SAS Institute Inc., Cary, NC, USA) as this is considered to be sufficiently robust for unbalanced data [41] and adjusted for multiplicity using Bonferroni. Salivary flow rates were analysed using explanatory variables of age, sex and with volunteer code as a random effect, and the dependent variable was saliva flow. The data relating to volunteers’ individual perception, liking and appetite was analysed using explanatory variables of age, sex, sample, saliva flow, mouth behaviour, dental status, medication, with volunteer code as a random effect. The dependent variables were (i) in the pilot study: attribute perception, liking, and JAR rating scores, and (ii) in the main study: attribute perception, liking and appetite ratings. Attribute data collected in the pilot study on the gLMS log-scale was first transformed to linear data (anti-logged). Values were expressed as least square means (LSM) estimates, as these values best reflect the statistical model. Penalty analysis was carried out by XLSTAT using cake and biscuit JAR and liking scores, with 20% selected as the threshold for population size. Penalty analysis evaluated the influence of volunteer perception of appropriateness of attribute level rating (JAR) on volunteer liking by calculating the mean drop in liking rating (scale 1–9) compared with mean liking of volunteers that rated the attribute as JAR (JAR 3 on 1–5 scale), determining whether this drop in liking score was significant. Analysis of significant preferences between cake and biscuit samples was calculated using a Binomial expansion in V-Power [42]. In all analysis *p* < 0.05 was used as the value for significant difference.

## 3. Results

### 3.1. Sensory Profile and Physical Properties of Cakes and Biscuits

QDA evaluation identified that 26 of the 34 attributes were significantly different between the control and protein cake and 10 of the 33 attributes were significantly different between the control and protein biscuit; demonstrating in both cases that protein fortification significantly increased “off” flavours (i.e., rancid or sulfurous), coupled either with increased mouthdrying or with a slower melting rate, when compared with the control versions. There was a similar result for cupcakes where 19 of the 36 attributes were significantly different between products. Although the introduction of lemon zest led to a reduction in “off” flavours compared with the pilot study cakes, the protein fortification still resulted in significantly increased mouthdrying and firmness of bite. Key attributes are summarised in Figure 3 (all attributes are outlined in Table 2 and Table 3). Protein fortification led to cakes and cupcakes that were perceived as substantially less dark in their yellow, whereas there was no significant difference in the perceived colour of the biscuits.

There were significant differences in physical properties between the control and protein versions of cakes and biscuits, as outlined in Table 4. Protein fortification of cakes and biscuits resulted in significantly reduced moisture content and significantly increased hardness, when compared with the control versions. Protein fortification of cupcakes resulted in no significant differences in moisture content between the two cupcakes; this was considered to be as a result of the improved balance of ingredients and was in contrast to the pilot study cakes. However, the protein cupcake did result in significant increases in height, hardness, cohesiveness, resilience and chewiness, when compared with the control cupcake. The protein fortified biscuits were found to be significantly darker, redder and more yellow in instrumental measurements. Colour differences between control and protein fortified cakes were less apparent, however, as with biscuits there was an overall colour difference of greater than 3, the minimum expected to lead to a perceptual difference [31]. These results do not completely parallel the sensory results (Figure 3) where protein fortification led to a considerably more noticeable colour difference (increased darkness) in cakes and cupcakes. However, the cakes are aerated and translucent which makes accurate instrumental colour measurements more difficult in comparison with the opaque biscuits. In conclusion, protein fortification led to increased colour development; however, this is less well represented by the instrumental readings of the cakes, due to aeration.

### 3.2. Dental Status and Mouth Behaviour Questionnaire Data

In both studies, the dental status of the volunteers was categorised into two groups: good dental status (20 or more teeth, no dentures and minimal missing teeth < 4) or reduced dental status (less than 20 teeth, dentures and missing teeth > 4). There was a significant association (*p* < 0.0001) between dental status and age in both studies, where predominately only older adults supported reduced dental status (Table 5). The mouth behaviour of the volunteers, as defined by Jeltema et al. [21,22], was classified into three types: chewers, crunchers and other/“smooshers” (pilot study: “smooshers” and suckers grouped together due to limited numbers in each group; main study: no suckers were recorded). Mouth behaviour was shown to be marginally independent (*p* = 0.06) of age in the pilot study and independent (*p* = 0.86) of age in the main study, where volunteers categorised themselves as chewers and crunchers more commonly than “smooshers”/other. All data is summarised in Table 5.

### 3.3. Salivary Flow Rates

Older adults demonstrated significantly lower unstimulated saliva flow (*p* < 0.05) compared with younger adults in both studies. However, age had no significant effect on stimulated saliva flow (Figure 4). Volunteers were grouped by tertile analysis into low, medium and high salivary flow rates, based on average unstimulated salivary flow rates. In the pilot study, there was a significant association (*p* = 0.01) between age and saliva flow grouping for unstimulated saliva. However, in the main study unstimulated saliva flow groupings were shown to be marginally insignificantly (*p* = 0.07) related to age (Table 5). Age was significantly associated (*p* < 0.0001) with medication, where only older adults reported regular medication use (Table 5). The effect of medication status on unstimulated saliva flow in older adults varied between the studies; in the pilot study there was no significant effect (*p* = 0.70), whereas in the main study there was a significant effect (*p* = 0.004) (Appendix A). The difference between the pilot and main studies may have been due to increasing experience with the saliva collection method (Appendix A) and an imbalance of the proportion of volunteers taking medication between the two studies. However, medication status had no significant effect on stimulated saliva flow in older adults (Appendix A). Dental status had no significant effect on unstimulated saliva flow; however, those volunteers with a reduced dental status had significantly (*p* < 0.05) lower stimulated saliva flow in both studies (Appendix A). Sex had a significant effect (*p* < 0.05) on saliva flow regardless of collection method, males having significantly higher salivary flow compared with females (Figure 4).

### 3.4. Individual Product Perception and Liking

During the pilot study, volunteers consumed a single bite of cake or biscuit, whereas during the main study volunteers consumed a full portion (35 g cupcake) at home. As detailed in Table 6 and Figure 5a and Figure 6, protein fortification of cakes and cupcakes significantly reduced (*p* < 0.05) overall liking, easiness to eat and swallow, sweetness and moistness and significantly increased (*p* < 0.05) mouthdrying, when compared with the control versions. Protein fortification of biscuits significantly reduced (*p* < 0.05) liking and moistness and significantly increased (*p* < 0.05) mouthdrying and hardness compared with the control biscuits. Regarding volunteer optimum levels for attributes (JAR scales, Table 7), protein fortification of cakes significantly reduced (*p* < 0.0001) flavour intensity and colour to below the optimum (3 = JAR), but significantly increased (*p* < 0.0001) biscuit colour to optimum.

During the pilot study, age significantly influenced (*p* < 0.05) liking and appropriateness of attributes. Older adults reported the protein cakes to be too low in flavour and colour, and biscuits to have a colour closer to optimum, when compared with the younger adults (Table 7). There were no further overall significant effects of age reported on liking, perception and easiness to consume. However, pairwise comparisons revealed that older adults found all protein fortified products (cakes, biscuits and cupcakes) to be significantly (*p* < 0.05) more mouthdrying compared with the control versions, whereas for younger adults this was only significant for cakes (Figure 5a,b and Figure 6).

There was no overall significant effect of saliva flow on liking, perception, ease to eat and swallow and JAR attributes; however, by categorising volunteers by unstimulated saliva flow, some trends did emerge. For biscuits, perceived mouthdrying intensity decreased with increasing salivary flow rates (low versus medium and high SF: *p* = 0.24 and *p* = 0.58, respectively). Whereas for cupcakes, perceived mouthdrying intensity increased with increasing salivary flow rates, regardless of the cupcake consumed (low versus medium and high SF: *p* = 0.30 and *p* = 0.37, respectively) (Appendix A).

In the pilot study, mouth behaviour type significantly influenced liking of appearance scores for cakes (*p* = 0.05) and biscuits (*p* = 0.009), where chewers gave higher scores than crunchers. Volunteers evaluated the same cakes at both study visits and scoring remained consistent in most cases (no significant effect of visit on cake liking, perception, JAR and preference) apart from easiness to eat/swallow, which were both rated as significantly less easy to eat (*p* = 0.01) and swallow (*p* = 0.002) at visit 2 (difference between visits: 0.2 on 5-point hedonic scale; Appendix A). In the main study, dental status significantly influenced (*p* < 0.05) liking and easiness to eat and swallow scores, where those with reduced dental status reported significantly lower scores compared with those with good dental status. No further significant effects were reported relating to mouth behaviour, dental status, medication and sex (Appendix A).

### 3.5. Preference and Penalty Analysis (Pilot Study Only)

There was a significant preference reported for the control cake (*p* < 0.0001) and control biscuit (*p* = 0.02) compared with the protein cake and biscuit (Appendix A). The volunteers’ perception of appropriateness of attribute level (JAR ratings) influenced their overall liking, as shown in the penalty analysis (Table 7). There was a significant influence of flavour, with older adults reporting ‘too little’ flavour more commonly than younger adults, which significantly penalised the liking scores.

### 3.6. Qualitative Feedback

In both studies volunteers’ comments were categorised into key themes, such as, flavour, texture, positive and negative comments and no comments provided (Appendix A). During the pilot study, there was general positive feedback provided for the control cake (186 positive comments out of 229 comments provided) and biscuits (72 positive comments out of 108 comments provided), demonstrating suitability for older adults. In contrast, negative comments were associated with the protein cake (175 negative comments out of 236 comments provided) and biscuits (79 negative comments out of 106 comments provided) relating to both flavour and texture, both of which were considered to be less appetising. During the main study, there was a general trend towards positive comments relating to the lemon flavour of both the control (59 positive comments out of 63 comments provided) and protein cupcakes (40 positive comments out of 63 comments provided). However, protein fortification of cupcakes (42 negative comments out of 62 comments provided) still resulted in a greater number of negative comments relating to texture, when compared with the control version (28 negative comments out of 62 comments provided). Examples of comments are summarised in Table 8.

### 3.7. Appetite Ratings (Main Study Only)

Consuming a 35 g protein cupcake significantly increased (*p* = 0.04) thirst compared with consumption of the same size control cupcake. Older adults reported that consumption of the protein cupcake significantly reduced (*p* = 0.02) prospective consumption, whereas younger adults did not. No further significant differences in appetite ratings were reported due to cupcake type or age. However, desire to eat ratings were significantly influenced by sex (*p* = 0.01), where females reported a greater reduction in desire to eat scores following cupcake consumption compared with males. There was also a significant effect of mouth behaviour (*p* = 0.03), where crunchers reported a lower reduction in desire to eat scores post consumption compared with “smooshers” and chewers. Two appetite ratings (desire to eat and prospective consumption) were significantly influenced (*p* < 0.05) by dental status, with a lower reduction in ratings following consumption from those with a reduced dental status, compared with those with good dental status. Unstimulated salivary flow rates were grouped using tertile analysis; however, despite some significant effects between samples and saliva flow groupings there were no clear trends (Table 9).

## 4. Discussion

### 4.1. Sensory Profile, Physical Properties, Perception and Liking of Products

Protein fortification of cakes and biscuits was associated with two key sensorial issues, namely mouthdrying and flavour, as well as with reduced acceptability and liking of products, when compared with the control versions. Milk proteins are considered to contribute to viscosity, structure and mouthfeel of dairy products and peptides and amino acids provide bases for volatile aroma active compounds [14]. Taste can be negatively impacted by aromatic amino acids, particularly when using high quality protein powders, and this provides additional challenges when increasing protein content in products [43]. However, fortifying foods with whey protein has been successfully achieved with positive results, for example, fortifying tomato sauces with WPI resulted in increased liking, when compared with the control, in healthy older adults [44].

The negative impact of protein fortification is likely to be explained by the differences in physical properties following fortification, such differences having been previously identified [45,46]. In the pilot study protein fortification led to a significant reduction in product moisture content when compared with control; this was addressed in the main study where the control and protein cupcake formulations were adjusted in order to achieve the same post-baking moisture content. It can therefore be assumed that any differences in perception and liking were not related to differences in final moisture content. In addition, water activity in the cakes was broadly similar, though the protein fortification of biscuits did significantly reduce water activity. Despite this, volunteers rated all protein fortified products to be significantly less moist and to cause more mouthdrying, in addition to the lingering mouthdrying evident from the sensory profile. Protein biscuits were also perceived to be significantly harder; this was supported by the texture analyser results which showed significantly increased hardness and fracturability scores. Mouthfeel attributes from the sensory profile, such as chewiness and firmness of bite, were significantly higher in the protein cakes, again supported by results from the texture analyser. Both the sensory profiling panel and the volunteers found the protein fortified products to be less sweet, the trained panel additionally concluding that vanilla and lemon flavours were reduced. Protein fortification led to cakes and cupcakes that were substantially darker in their yellow colour (as rated by the sensory panel) and biscuits that were significantly darker and redder (as measured instrumentally). However, the colour differences between the control and protein versions had no effect on the mean liking of product appearance. These results are generally supported by previous studies; for example, Tsikritzi et al. noted that whey protein fortification can result in undesirable texture and flavours [28,44].

There was evidence of Maillard reactions which can influence the colour and flavour of products [45], as such protein fortification of cakes and biscuits (products low in moisture and water activity content) led to more colour development (darker and redder) from increased Maillard reactions due to the increase in available amino groups. In addition, there are differences in water absorption between flour and dairy powders, implying less water activity available, further contributing to more optimum conditions for the Maillard reaction [45]. The rate of Maillard reaction is considered dependent on water activity, temperature and level of precursors (i.e., reducing sugars and available amino acid groups) [13]. Furthermore, within the protein cakes, it is likely that WPI/WPC contributed to more air bubbles being trapped within the batter, accordingly increasing the cake volume [46]. Whey proteins can also become unstable when heated, resulting in protein denaturation and aggregation, influencing both the structure and the stability of the protein [47]. Denaturation of proteins during the baking process is considered to influence the protein’s interactions, elasticity, binding sites and flavour, leading to changes in the final product [14,46,48].

Proteins are reported to interact with flavour compounds by releasing and binding to them and thereby changing perception of the product flavour [49]. This could explain the blunting and reduced intensity of flavour perceived by both the trained panel and the volunteers, as well as the number of comments made as to blandness and lack of flavour from the volunteers. In addition, whey proteins are rich in sulfur amino acids and upon heating can release sulfurous and eggy aromas which influence overall flavour [13,14]. These effects could potentially explain the “off” flavours, such as, eggy, rancid, fatty and sulfate flavours, identified by the trained panel after consumption of the protein products, and therefore are potential contributors to the reduced acceptability of products. It should be noted that these “off” flavours were more evident in the cakes and biscuit in the pilot study, though still evident in the protein cupcakes in the main study, but to a lesser extent. It is therefore likely that the addition of lemon zest had a positive impact on both flavour and acceptability in the main study compared with the cakes used in the pilot study. Future research needs to focus on understanding the causes of “off” flavours, as well as texture differences between the control and protein versions, and how these influence the acceptability of the final product.

### 4.2. Individual Differences in Perception and Liking of Products

Changes in oral impairments and sensory sensitivity are commonly associated with ageing and are considered to influence food consumption [2]. Our study demonstrated that individuals with reduced dental status reported significantly lower liking and easiness to eat and swallow scores in the main study only, where predominately only older adults had reduced dental status. Accordingly, these findings are supportive of developing foods appropriate for older adults, for example those with reduced dental status. Jeltema et al. proposed that mouth behaviour can also influence food choice, texture preference and satisfaction [21,22]. However, both studies demonstrated that individual differences in mouth behaviour had no effect on volunteers’ ratings of products, apart from appearance scores for cakes and biscuits in the pilot study. Therefore, regardless of mouth behaviour type it can be assumed that volunteers rate perception and liking to the same extent. The influence of age was less clear on individual perception and liking, for example, there was no overall effect of age. However, older adults perceived the protein versions, regardless of the food type, to cause more mouthdrying compared with the control versions. This was despite using two different methods to measure perception and liking: in the pilot study using a single point in time and in the main study using a full portion size at home. These findings did support previous work in a liquid model system, where older adults were shown to have a greater sensitivity to mouthdrying [19].

Our study showed that older adults demonstrated a significantly lower unstimulated salivary flow rate, when compared with younger adults, supporting the findings of Vandenberghe-Descamps et al. that salivary flow rates decrease with age [25]. However, our study demonstrated no age-related differences in stimulated salivary flow rates, which was consistent with the findings in our previous work [19]. Furthermore, it is proposed that age-related changes in salivary flow rates are gland specific, with the parotid and minor salivary glands potentially being less influenced by age [50]. Saliva plays a key role within our eating experience [51] and accordingly it was expected that a reduced saliva flow could influence perception and liking of products. However, the trends proved inconsistent and varied depending on the product type, and typically volunteers perceived the differences to the same extent despite their differences in saliva flow. For example, as expected, perceived mouthdrying intensity decreased with increasing salivary flow rates with the protein biscuits, which are a harder, drier and a less moist product. It is recognised that saliva performs a key role in ensuring that the food bolus is moist and lubricated so that the product can be safety consumed [52]. Reduced saliva flow is associated with poor oral clearance [53] and accordingly food particles are more likely to linger within the mouth, thereby increasing perception of mouthdrying in relation to the biscuits. However, perceived mouthdrying intensity increased with increasing salivary flow rates following the protein cupcake; this was a surprisingly trend, given that cupcakes are a softer and smoother product. This latter finding is, however, supported by our previous work in a whey protein beverage model [19] where we suggested a potential hydration mechanism associated with muco-adhesion, with the lubrication abilities of saliva strengthening adhesion properties and resulting in an increased perception of mouthdrying [54]. The results of the current study, which were different between biscuits and cakes, might indicate that the effect of saliva flow on the perception of mouthdrying, and the underlying mechanism, will be dependent on factors in addition to protein content, such as the structure and moisture content of the food. For example, the role of food particle size needs to be considered and understood [55]. Within a solid system, such as cakes and biscuits, saliva plays a role in determining whether a food particle will aggregate or adhere to the oral mucosa, with the latter increasing friction and influencing sensory perception [56]. It is proposed that muco-adhesion strength should be greater within a solid model, when compared with a liquid system [54].

Given the derived benefits from ONS and protein fortified products, it is important to understand the causes of poor compliance and consumer acceptance, which have been associated with mouthdrying and “off” flavours. Accordingly, appetite ratings, such as how hunger and thirst influence sensory perception and consumption of products, is of particular relevance [26]. Thomas et al. demonstrated that multi sips of ONS can increase thirst during consumption and also correlated this with increased drying sensations [26]; our study also demonstrated that protein cupcakes increased perception of mouthdrying and thirst.

### 4.3. Limitations

The main limitations of this study relate to sex and health. First, the study identified sex related effects; however, it should be noted that the study had a sex imbalance, particularly in the younger adults. Second, studies with older adults should have sufficient sample size to allow for the diverse nature of older adults within a group described as ‘healthy’ and to ensure sufficient power between age groups [57]. Despite aiming for healthy older adults with minimal medication use, the studies presented in this paper include volunteers of differing age, medication use, level of impairment (physical, visual and hearing) and previous experience of sensory studies. All these additional factors are likely to have influenced individual perception of a product, in addition to any salivary flow changes associated with age.

## 5. Conclusions

Protein fortification of cakes and biscuits significantly increased perceived mouthdrying, hardness and “off” flavours and significantly reduced melting rate, moistness and liking, when compared with the control versions. Such intensity and direction of attributes are likely to have contributed to dislike of and poor compliance with products and indicate the need for reformulation of the products to ensure product suitability for older adults. Consumption of simple and familiar snacks, such as cakes and biscuits, can help to alleviate malnutrition and sarcopenia, however they clearly need to be acceptable and palatable. Individual differences (such as age, mouth behaviour, dental status, saliva flow and appetite) were expected to play a greater role in perception and liking of products; trends were, however, present but inconclusive, indicating the need for further research into the impact of these individual differences. Further investigation remains necessary to understand the causes of mouthdrying resulting from solid food models and to establish whether there is a link between mouthdrying and muco-adhesion. In addition, there would be a clear benefit for older adults at risk of malnutrition and sarcopenia were these effects (mouth drying and “off” flavours) to be mitigated.

## Figures and Tables

**Figure 1 foods-09-01328-f001:**
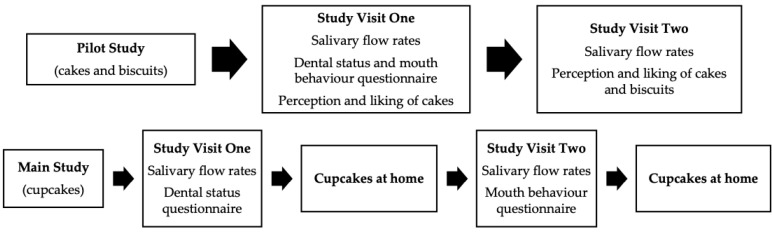
Overview of pilot and main studies.

**Figure 2 foods-09-01328-f002:**
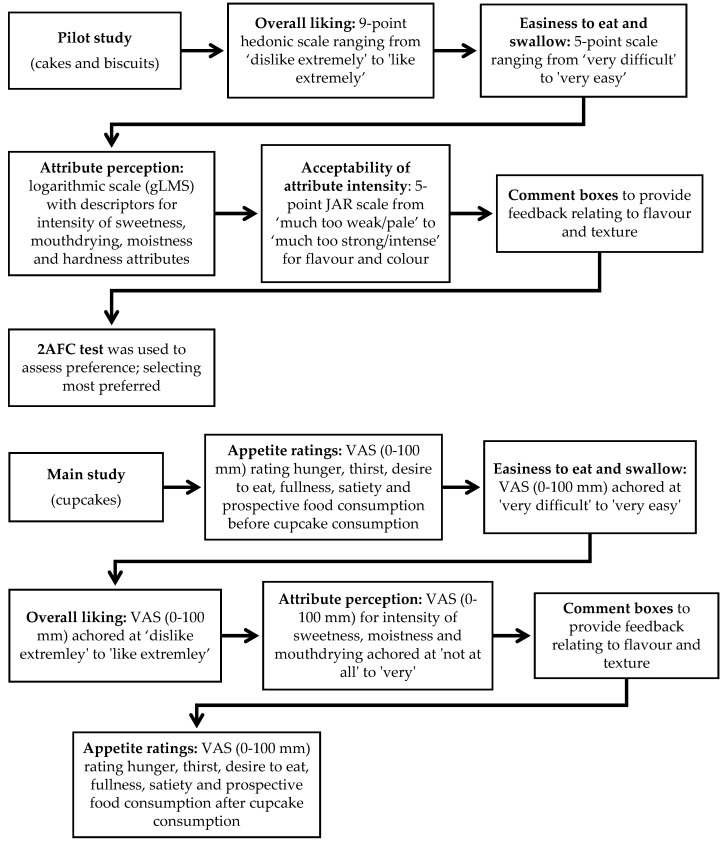
Overview of individual product perception and liking measures taken during both studies (GLMS: generalised Labelled Magnitude Scale; JAR: Just-About-Right; 2AFC: two alternative forced choice; VAS: visual analogue scale).

**Figure 3 foods-09-01328-f003:**
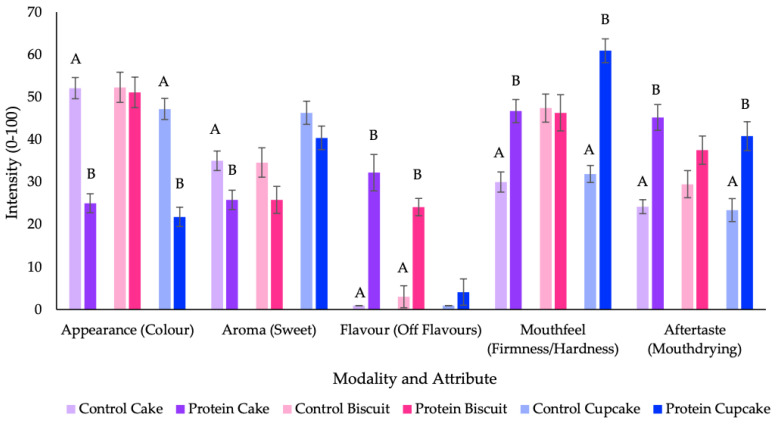
Summary of key sensory profile attributes for each modality and product (measured on a visual analogue scale (VAS), 0–100). Values are expressed as mean ± standard error. Significant differences (*p* < 0.05) between control and protein sample pairs are denoted by differing letters; no letter between sample pair reflects no significant difference. Colour (cake/cupcake: darkness of yellow colour of crumb; biscuit: darkness of colour (top)) and firmness refers to cakes/cupcakes and hardness refers to biscuits.

**Figure 4 foods-09-01328-f004:**
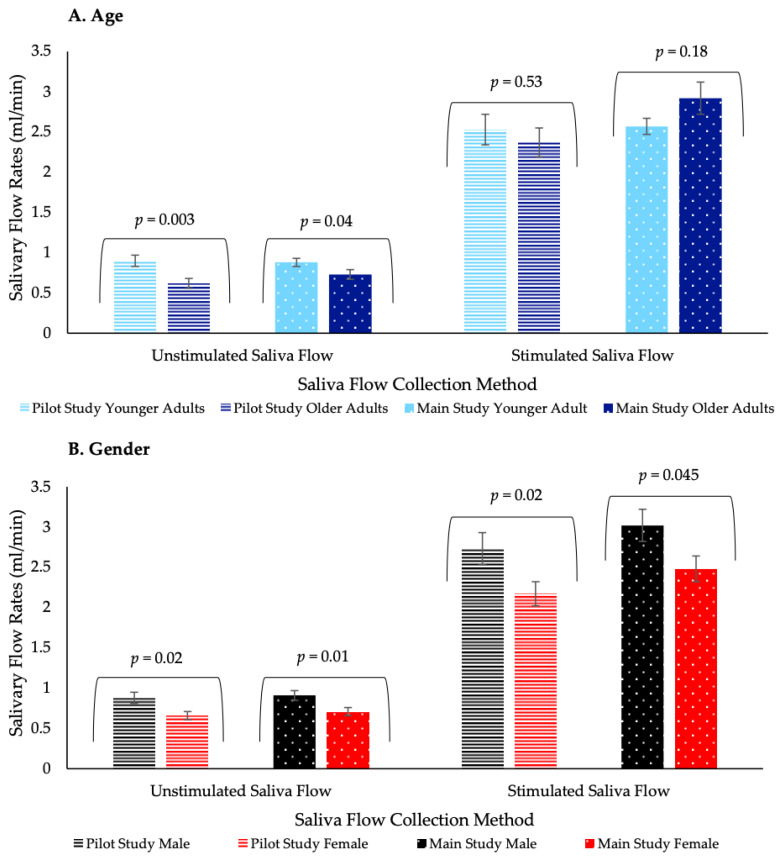
Summary of volunteers’ salivary flow rates (mL/min) in both studies. Values are expressed as least square means (LSM) estimates ± standard error from SAS output. Significant differences (*p* < 0.05) are reported between groups with relevant *p* value above each group.

**Figure 5 foods-09-01328-f005:**
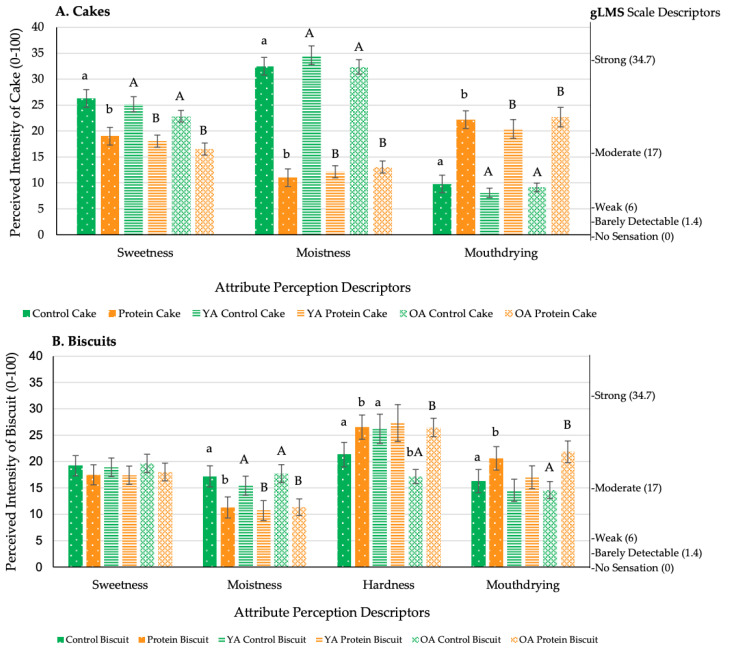
Volunteers attribute perception mean ratings of (**A**) cakes and (**B**) biscuits by overall and age (pilot study: *n* = 84; generalised Labelled Magnitude Scale (gLMS) anti-logged data, scale 0–100 summarised on the right of the figure). Values are expressed as LSM estimates ± standard error from SAS output. Significant differences (*p* < 0.05) between samples (by overall: control vs. protein, and age: younger adults (YA) vs. older adults (OA)) are denoted by differing small letters, and significant differences within age groups are denoted by differing capital letters; no letter reflects no significant difference.

**Figure 6 foods-09-01328-f006:**
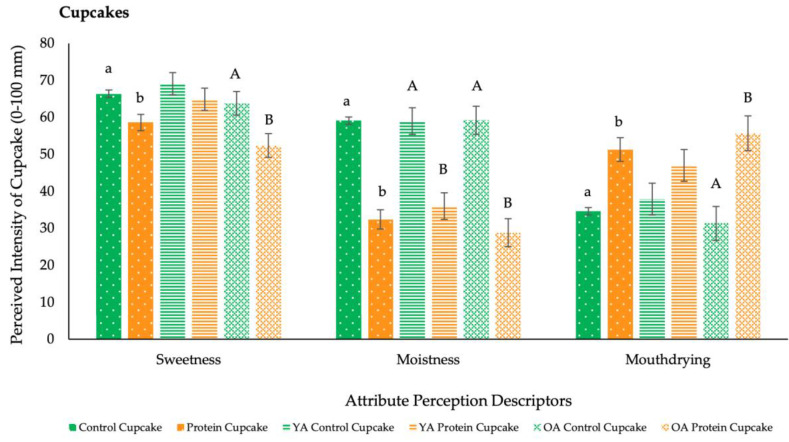
Volunteers attribute perception mean ratings of cupcakes by overall and age (main study: *n* = 70; VAS, 0–100 mm). Values are expressed as LSM estimates ± standard error from SAS output. Significant differences (*p* < 0.05) between samples (by overall: control vs. protein, and age: younger adults (YA) vs. older adults (OA)) are denoted by differing small letters, and significant differences within age groups are denoted by differing capital letters; no letter reflects no significant difference.

**Table 1 foods-09-01328-t001:** Nutritional composition per portion size and 100 g with % of reference intake for cakes, biscuits and cupcakes.

	Control Cake			Protein Cake			Control Biscuit			Protein Biscuit			Control Cupcake			Protein Cupcake		
	per 45 g Slice	per 100 g	%	per 45 g Slice	per 100 g	%	per 20 g Biscuit	per 100 g	%	per 20 g Biscuit	per 100 g	%	per 35 g Cupcake	per 100 g	%	per 35 g Cupcake	per 100 g	%
Energy (kcal)	184	410	21	185	411	21	119	597	30	118	588	29	155	442	22	156	445	22
Fat (g)	10	23	33	9.8	22	31	7.2	36	51	6.7	34	49	8.2	24	34	8.4	24	34
of which saturates (g)	6.2	14	70	5.8	13	65	0.5	2.7	14	0.5	2.5	13	4.9	14	70	5	14	70
Carbohydrate (g)	20	44	17	19	41	16	12	61	23	11	56	22	18	51	20	16	45	17
of which sugars (g)	12	26	29	11	24	27	5.2	26	29	4.8	24	27	9.2	26	29	9.3	26	29
Fibre (g)	0.6	1.4	6	0.6	1.3	5	0.7	3.4	14	0.6	3.1	12	0.5	1.1	4	0.5	1.1	4
Protein (g)	2.7	6	12	5.3	12	24	1.1	5.6	11	2.7	14	28	2.1	6	12	4.1	12	24
Salt (g)	0.3	0.7	12	0.3	0.7	12	0.06	0.3	5	0.07	0.4	6	0.1	0.3	5	0.1	0.3	5

**Table 2 foods-09-01328-t002:** Summary of sensory profile with corresponding reference/description and results for cakes and cupcakes. Values are expressed as means (measured on a visual analogue scale (VAS, 0–100)).

Modality	Attribute	Reference and/or Description	Control Cake	Protein Cake	Significance of Sample (*p*-Value)	Control Cupcake	Protein Cupcake	Significance of Sample (*p*-Value)
Appearance	Moist appearance	Slightly or moderately wet to touch	50.4	23.6	**0.0002**	48.5	15.7	**<0.0001**
	Dense appearance of sponge	Compact in structure	51.9	37.9	0.006	31.6	64.2	**<0.0001**
	Appearance of large holes in sponge	Holes within crumb structure	12.5	22.9	**0.03**	25.3	48.4	**0.0009**
	Yellow colour of crumb (inside)	Intensity of yellow colour within crumb	52.1	25.0	**<0.0001**	47.2	21.8	**<0.0001**
Aroma	Overall odour intensity	Intensity of aroma within cake	47.8	50.9	0.43	54.2	48.8	0.16
	Sweet	Sucrose (5.76 g/L)	35.0	25.8	**0.01**	46.3	40.4	0.17
	Vanilla/lemon	Vanilla extract/lemon zest	21.8	10.7	**0.009**	44.5	36.5	0.08
	Buttery	Cooked butter (melted unsalted butter)	20.8	8.9	**0.0001**	23.0	6.9	**0.002**
	Eggy	Intensity of eggy note	36.2	9.5	**<0.0001**	7.8	17.1	**0.005**
	Rancid/off flavours	Curded buttermilk (cooked buttermilk)	0.0	32.9	**<0.0001**	0.0	6.2	0.11
Flavour	Overall flavour intensity	Intensity of flavour within cake	48.9	49.9	0.82	51.2	40.3	**0.03**
	Sweet	Sucrose (5.76 g/L)	40.7	27.0	**0.002**	46.8	40.0	0.19
	Metallic	Iron (II) sulphate heptahydrate (0.0036 g/L)	1.8	1.2	0.64	2.1	4.5	0.20
	Vanilla/lemon	Vanilla extract/lemon zest	25.0	11.1	**0.0009**	44.2	33.0	**0.02**
	Buttery	Cooked butter (melted unsalted butter)	22.0	8.0	**0.001**	20.8	6.3	**0.03**
	Eggy	Intensity of eggy note	31.3	8.9	**<0.0001**	7.0	12.4	**0.03**
	Liquorice	Liquorice (liquorice twists)	n/a	n/a	n/a	0.9	5.7	0.053
	Rancid/off flavours	Curded buttermilk (cooked buttermilk)	0.0	32.3	**<0.0001**	0.0	4.1	0.22
Mouthfeel	Firmness of bite	Degree of force with first bite	30.0	46.7	**0.001**	31.9	60.9	**<0.0001**
	Moist sponge	Slightly damp sponge	59.1	15.9	**<0.0001**	55.1	16.5	**<0.0001**
	Chewy	Ease of ability to chew	11.9	46.4	**0.0001**	29.0	59.6	**<0.0001**
	Greasy lips	Degree of oiliness/greasiness on lips	15.7	7.2	**0.048**	14.5	2.3	**0.02**
	Crumbliness of sponge	Ease to break into small pieces	30.3	41.4	0.18	31.7	23.3	0.22
	Crumb size	Size of crumb inside of cake	27.7	52.2	**0.0004**	23.4	45.4	**0.0002**
	Pasty (cohesive)	Sticking to surfaces	27.0	9.1	**0.014**	46.4	17.2	**0.0001**
	Rate of breakdown & clearance	Clearing sample from mouth	52.8	31.5	**0.0014**	55.9	28.8	**0.0006**
	Cooling sensation (numbing)	A stimulation resulting in feeling of coolness	6.8	3.4	0.20	1.1	4.3	0.22
Aftertaste	Mouthdrying	Drying sensation in the mouth	24.2	45.2	**0.0001**	23.4	40.8	**0.001**
	Sweet	Sucrose (5.76 g/L)	33.6	21.6	**0.0002**	40.5	36.7	0.38
	Vanilla/lemon	Vanilla extract/lemon zest	17.9	6.1	**0.006**	33.3	21.2	**0.007**
	Buttery	Cooked butter (melted unsalted butter)	18.5	2.8	**0.0001**	11.3	3.6	**0.09**
	Rancid/off flavours	Curded buttermilk (cooked buttermilk)	0.0	18.8	**0.0008**	0.0	2.3	0.17
	Salty	Sodium chloride (1.19 g/L)	1.3	2.5	0.10	2.6	1.7	0.42
	Salivating	Increased saliva within mouth	26.9	31.5	0.08	29.7	29.0	0.84
	Metallic	Iron (II) sulphate heptahydrate (0.0036 g/L)	6.8	7.5	0.86	2.7	8.1	0.07
	Liquorice	Liquorice (liquorice twists)	n/a	n/a	n/a	1.0	5.8	0.06

n/a refers to not applicable.

**Table 3 foods-09-01328-t003:** Summary of sensory profile with corresponding reference/description and results for biscuits. Values are expressed as means (measured on a visual analogue scale (VAS, 0–100)).

Modality	Attribute	Reference and/or Description	Control Biscuit	Protein Biscuit	Significance of Sample (*p*-Value)
Appearance	Evenness of shape	Uniform shape of biscuit	51.4	58.0	0.08
	Smoothness of surface	Texture without roughness, top surface only	31.2	35.4	0.21
	Darkness of colour (top of biscuit)	Intensity of colour, top surface only	52.3	51.1	0.79
	Darkness of colour (inside of biscuit)	Intensity of colour, inside surface only	45.9	32.3	**0.0001**
	Darkness of colour (bottom of biscuit)	Intensity of colour, bottom surface only	73.5	79.5	**0.007**
	Thickness	Degree of thickness of biscuit	43.6	54.0	**0.006**
	Crumb/aeration	Size of crumb biscuit	38.4	42.3	0.27
Aroma	Lemon	Lemon (sliced lemon)	35.2	33.2	0.71
	Sweet	Sucrose (5.76 g/L)	34.6	25.8	0.06
	Oaty	Raw oats	18.4	15.2	0.32
	Fatty	Piece of lard	9.6	8.2	0.75
	Baked	Cooked in an oven	48.9	41.0	0.09
	Sulfate off note	Cooked cabbage	4.9	18.1	0.08
Flavour	Sweet	Sucrose (5.76 g/L)	43.7	28.8	**0.02**
	Oaty	Raw oats	25.6	23.6	0.46
	Fatty	Piece of lard	8.6	14.5	0.08
	Bitter	Quinine (0.04 g/L)	9.7	11.8	0.45
	Lemony	Lemon zest	33.1	23.8	**0.047**
	Sulfate off note	Cooked cabbage	3.0	24.1	**0.002**
	Metallic	Iron (II) sulphate heptahydrate (0.0036 g/L)	3.7	6.0	0.06
Mouthfeel	Hardness of first bite	Degree of force with first bite	47.4	46.3	0.72
	Crumbly	Ease of break into small pieces	59.6	50.4	0.06
	Melt Rate/ dissolving rate	Speed of dissolve and melt within mouth	56.7	48.3	**0.004**
	Mouthdrying	Drying sensation in the mouth	35.5	42.8	0.19
	Teeth packing	Biscuit sticking to the surface of teeth	58.3	50.7	**0.03**
	Grainy	Not smooth or fine, rough to touch	39.3	35.1	0.38
	Crunchy	Degree of force and sound with chewing	45.7	41.7	0.33
Aftertaste	Sweet	Sucrose (5.76 g/L)	28.9	18.3	**0.02**
	Teeth packing (residue)	Biscuit sticking to the surface of teeth	44.6	41.2	0.47
	Mouthdrying	Drying sensation in the mouth	29.5	37.5	0.14
	Lemony	Lemon zest	22.4	15.9	0.06
	Bitter	Quinine (0.04 g/L)	10.2	10.6	0.79
	Sulfate off note	Cooked cabbage	0.8	10.0	**0.003**

**Table 4 foods-09-01328-t004:** Physical properties of cakes, biscuits and cupcakes.

Measure	Control Cake	Protein Cake	Control Biscuit	Protein Biscuit	Control Cupcake	Protein Cupcake
Height (mm)	60.6 ± 1.0 ^a^	70.5 ± 1.2 ^b^	-	-	28.0 ± 0.5 ^a^	41.3 ± 0.6 ^b^
Diameter (mm)	-	-	61.9 ± 0.4	62.7 ± 0.2	-	-
Moisture Content (%)	28.8 ± 0.4 ^a^	25.3 ± 0.5 ^b^	3.1 ± 0.04 ^a^	2.3 ± 0.05 ^b^	23.5 ± 0.4	24.2 ± 0.2
Water Activity (aw)	0.87 ± 0.003	0.87 ± 0.003	0.36 ± 0.01 ^a^	0.27 ± 0.007 ^b^	0.81 ± 0.0005	0.85 ± 0.004
Colour *L** (light-dark)	74.0 ± 3.8	73.2 ± 2.1	61.1 ± 2.4 ^a^	58.3 ± 2.3 ^b^	78.7 ± 0.5	77.8 ± 1.0
Colour *a** (green-red)	−3.8 ± 0.2	−3.6 ± 0.2	3.9 ± 0.9 ^a^	10.4 ± 1.6 ^b^	−3.7 ± 0.1	−3.3 ± 0.03
Colour *b** (blue-yellow)	29.0 ± 1.9 ^a^	26.2 ± 0.3 ^b^	28.6 ± 1.2 ^a^	32.5 ± 1.7 ^b^	28.8 ± 0.1	25.6 ± 0.5
Hue Angle (arctan (*b**/*a**))	97.4 ± 0.2	97.8 ± 0.1	82.2 ± 0.4 ^a^	71.2 ± 1.1 ^b^	97.5 ± 0.4	97.5 ± 0.3
Colour Difference (dE)	3.8 ± 0.7		8.9 ± 0.6		3.6 ± 0.5	
Hardness (g)	181 ± 18 ^a^	372 ± 18 ^b^	2384 ± 140 ^a^	3201 ± 178 ^b^	784 ± 50 ^a^	1130 ± 83 ^b^
Chewiness (-)	135 ± 48 ^a^	294 ± 53 ^b^	-	-	504 ± 51 ^a^	814 ± 86 ^b^
Springiness (%)	0.96 ± 0.01	0.96 ± 0.01	-	-	89.5 ± 3.6	89.1 ± 1.8
Cohesiveness (-)	0.78 ± 0.01 ^a^	0.82 ± 0.009 ^b^	-	-	0.71 ± 0.03 ^a^	0.80 ± 0.02 ^b^
Adhesiveness (g.s)	-	-	-	-	−2.2 ± 0.8 ^a^	−0.06 ± 0.2 ^b^
Resilience (%)	-	-	-	-	29.2 ± 1.5	31.9 ± 0.8
Gumminess (-)	-	-	-	-	559 ± 41 ^a^	910 ± 82 ^b^
Fracturability (mm)	-	-	43.0 ± 6.7 ^a^	84.2 ± 12 ^b^	-	-

Values are expressed as mean of three replicates ± standard error. Significant differences (*p* < 0.05) between control and protein sample pairs are denoted by differing letters. (-) represents no unit (dimensionless data). Some measures did not apply to all product types; in the table above, the colour measurements relate to the crumb of cake and cupcake, and the top surface of biscuit.

**Table 5 foods-09-01328-t005:** Summary of volunteers’ sex, medication, dental status, mouth behaviour and unstimulated saliva flow groupings in both studies.

	Sex				Medication				Dental Status				Mouth Behaviour						Saliva Flow Categories					
	Male		Female		Yes		No		Good		Reduced		Chewer		Cruncher		Other		Low		Medium		High	
	*n*	%	*n*	%	*n*	%	*n*	%	*n*	%	*n*	%	*n*	%	*n*	%	*n*	%	*n*	%	*n*	%	*n*	%
**Pilot Study**																								
Total (*n* = 84)	31	37	53	63	19	23	65	77	64	76	20	24	42	50	33	39	9	11	27	32	28	33	29	35
YA (*n* = 42)	12	29	30	71	0	0	42	100	41	98	1	2	23	55	12	20	7	16	8	19	14	33	20	48
OA (*n* = 42)	19	45	23	55	19	45	23	55	23	55	19	45	19	45	21	50	2	5	19	45	14	33	9	21
**Main Study**																								
Total (*n* = 70)	27	39	43	61	18	26	52	74	59	84	11	16	29	43	25	37	13	19	23	33	23	33	24	34
YA (*n* = 38)	13	34	25	66	0	0	38	100	38	100	0	0	16	42	15	39	7	18	8	21	15	39	15	39
OA (*n* = 32)	14	44	18	56	18	56	14	44	21	66	11	34	13	45	10	34	6	21	15	47	8	25	9	28

‘*n*’ and ‘%’ reflect number and percentage in each contributing group. YA: younger adult and OA: older adult. Mouth behaviour ‘other’ reflects “smooshers”/sucker in the pilot study and “smooshers” in the main study. Missing data for mouth behaviour (main study *n* = 3). In both studies dental status is significantly associated with medication (pilot study: *p* = 0.006; main study: *p* < 0.0001) and is independent of mouth behaviour category (pilot study: *p* = 0.95; main study: *p* = 0.97). Saliva flow groupings are derived from unstimulated salivary flow and categories are defined by tertiles with mL/min range for each category (pilot study: low saliva flow: 0.04–0.53 mL/min; medium saliva flow: 0.53–0.77 mL/min; high saliva flow: 0.77–2.18 mL/min and main study: low saliva flow: 0.23–0.58 mL/min; medium saliva flow: 0.58–0.95 mL/min; high saliva flow: 0.95–1.52 mL/min).

**Table 6 foods-09-01328-t006:** Volunteers’ liking and easiness to eat and swallow mean ratings of cakes, biscuits and cupcakes in both studies; overall and by age and unstimulated saliva flow rate.

	Overall		Age		Unstimulated Saliva Flow		
		Significance of Sample (*p*-Value)	Younger Adults	Older Adults	Low Saliva Flow ^3^	Medium Saliva Flow ^4^	High Saliva Flow ^5^
**Appearance Liking**							
Control Cake ^1^	6.7 ± 0.1	0.20	6.6 ± 0.2	6.8 ± 0.2	6.7 ± 0.2	6.8 ± 0.2	6.6 ± 0.2
Protein Cake ^1^	6.9 ± 0.1		6.9 ± 0.3	6.9 ± 0.2	7.1 ± 0.2	6.8 ± 0.2	6.7 ± 0.2
Control Biscuit ^1^	5.6 ± 0.2	0.08	4.8 ± 0.4 ^aA^	6.4 ± 0.3 ^b^	5.4 ± 0.3	5.4 ± 0.3	5.8 ± 0.4
Protein Biscuit ^1^	5.9 ± 0.2		5.5 ± 0.4 ^B^	6.3 ± 0.3	5.5 ± 0.3	6.0 ± 0.3	6.2 ± 0.4
Control Cupcake ^2^	57.2 ± 3.8	0.18	59.8 ± 5.5	54.7 ± 4.5 ^A^	56.3 ± 5.4	58.0 ± 5.1	57.3 ± 5.8
Protein Cupcake ^2^	60.0 ± 3.8		56.2 ± 5.4	65.5 ± 4.4 ^B^	56.6 ± 5.2	61.4 ± 5.0	64.5 ± 5.8
**Overall Liking**							
Control Cake ^1^	6.6 ± 0.2	**<0.0001**	6.5 ± 0.3 ^A^	6.9 ± 0.3 ^A^	6.5 ± 0.3 ^A^	6.9 ± 0.3 ^A^	6.7 ± 0.3 ^A^
Protein Cake ^1^	5.0 ± 0.2		5.0 ± 0.3 ^B^	5.0 ± 0.2 ^B^	5.2 ± 0.3 ^B^	4.8 ± 0.3 ^B^	5.0 ± 0.3 ^B^
Control Biscuit ^1^	6.2 ± 0.2	**0.002**	5.5 ± 0.4 ^a^	7.0 ± 0.3 ^bA^	5.8 ± 0.4	6.5 ± 0.3 ^A^	6.4 ± 0.4
Protein Biscuit ^1^	5.3 ± 0.2		4.8 ± 0.4	5.9 ± 0.3 ^B^	5.0 ± 0.4	5.3 ± 0.3 ^B^	5.6 ± 0.4
Control Cupcake ^2^	65.4 ± 4.0	**<0.0001**	68.2 ± 5.9 ^A^	62.6 ± 4.8 ^A^	62.2 ± 5.8	68.8 ± 5.5 ^A^	65.0 ± 6.2 ^A^
Protein Cupcake ^2^	51.3 ± 4.0		56.7 ± 5.8 ^B^	45.8 ± 4.7 ^B^	53.8 ± 5.6	48.9 ± 5.4 ^B^	51.0 ± 6.3 ^B^
**Easiness to Eat**							
Control Cake ^1^	4.3 ± 0.1	**<0.0001**	4.2 ± 0.2 ^A^	4.3 ± 0.1 ^A^	4.2 ± 0.1 ^A^	4.1 ± 0.1 ^A^	4.3 ± 0.1 ^A^
Protein Cake ^1^	3.2 ± 0.1		3.2 ± 0.2 ^B^	3.1 ± 0.1 ^B^	3.3 ± 0.1 ^B^	3.1 ± 0.1 ^B^	3.1 ± 0.1 ^B^
Control Biscuit ^1^	3.5 ± 0.1	0.23	3.0 ± 0.2 ^a^	4.0 ± 0.2 ^bA^	3.3 ± 0.2	3.7 ± 0.2	3.5 ± 0.2
Protein Biscuit ^1^	3.3 ± 0.1		3.5 ± 0.2	3.5 ± 0.2 ^B^	3.4 ± 0.2	3.3 ± 0.2	3.3 ± 0.2
Control Cupcake ^2^	67.3 ± 3.9	**<0.0001**	66.8 ± 5.6 ^A^	67.8 ± 4.7 ^A^	68.7 ± 5.7 ^A^	65.1 ± 5.4 ^A^	68.0 ± 6.0 ^A^
Protein Cupcake ^2^	49.4 ± 3.9		50.0 ± 5.6 ^B^	48.8 ± 4.6 ^B^	57.9 ± 5.5 ^B^	46.9 ± 5.4 ^B^	43.5 ± 6.0 ^B^
**Easiness to Swallow**							
Control Cake ^1^	4.0 ± 0.1	**<0.0001**	3.9 ± 0.2 ^A^	4.1 ± 0.1 ^A^	4.0 ± 0.1 ^A^	3.9 ± 0.1 ^A^	4.1 ± 0.2 ^A^
Protein Cake ^1^	3.0 ± 0.1		2.9 ± 0.2 ^B^	3.1 ± 0.1 ^B^	3.1 ± 0.1 ^B^	2.9 ± 0.1 ^B^	2.9 ± 0.2 ^B^
Control Biscuit ^1^	3.4 ± 0.1	0.95	3.0 ± 0.2 ^a^	3.8 ± 0.2 ^b^	3.2 ± 0.2	3.5 ± 0.1	3.6 ± 0.2
Protein Biscuit ^1^	3.4 ± 0.1		3.3 ± 0.2	3.5 ± 0.2	3.5 ± 0.2	3.3 ± 0.1	3.5 ± 0.2
Control Cupcake ^2^	64.5 ± 3.7	**<0.0001**	62.4 ± 5.2 ^A^	66.5 ± 4.5 ^A^	65.4 ± 5.6	62.1 ± 5.3 ^A^	65.9 ± 5.7 ^A^
Protein Cupcake ^2^	47.7 ± 3.7		48.8 ± 5.2 ^B^	46.5 ± 4.5 ^B^	56.0 ± 5.3 ^a^	49.1 ± 5.2 ^aB^	37.9 ± 5.7 ^bB^

Values are expressed as LSM estimates ± standard error from SAS output. Significant differences (*p* < 0.05) within a row (i.e., age YA vs. OA and saliva flow pairwise comparisons) are denoted by differing small letters; and within a column (i.e., within an age group between samples or within saliva flow groupings between samples) are denoted by differing capital letters. During the pilot study ^1^ (*n* = 84; YA: *n* = 42; OA: *n* = 42) all cakes and biscuits were measured on a 9- and 5-point scale, respectively and during the main study ^2^ (*n* = 70; YA: *n* = 38; OA: *n* = 32) all cupcakes were measured on a visual analogue scale (VAS) 0–100 mm. Individual saliva flow groupings are derived from unstimulated saliva flow only, through tertile analysis (low saliva flow ^3^, pilot study *n* = 27; main study *n* = 23; medium saliva flow ^4^, pilot study *n* = 28; main study *n* = 23; high saliva flow ^5^, pilot study *n* = 29; main study *n* = 24).

**Table 7 foods-09-01328-t007:** Volunteers’ appropriateness of attribute level (Just-About-Right, JAR) mean ratings of cakes and biscuits and their influence on volunteer liking ratings in the pilot study; overall and by age (YA: younger adult and OA: older adult) and by unstimulated saliva flow rate.

	Overall (*n* = 84)		Age		Unstimulated Saliva Flow			Penalty Analysis (Mean Liking Drop Where Attribute Deviates from Just-About-Right)			
		Significance of Sample (*p*-Value)	Younger Adults (*n* = 42)	Older Adults (*n* = 42)	Low Saliva Flow (*n* = 27)	Medium Saliva Flow (*n* = 28)	High Saliva Flow (*n* = 29)	Too Little (YA)	Too Much (YA)	Too Little (OA)	Too Much (OA)
**JAR Flavour**											
Control Cake	2.9 ± 0.08	**<0.0001**	2.9 ± 0.1	2.8 ± 0.1 ^A^	2.7 ± 0.1 ^a^	3.0 ± 0.1 ^bA^	2.8 ± 0.1 ^aA^	0.65	1.68	1.79 ^#^	−0.70
Protein Cake	2.6 ± 0.08		2.8 ± 0.1 ^a^	2.3 ± 0.1 ^bB^	2.4 ± 0.1	2.6 ± 0.1 ^B^	2.5 ± 0.1 ^B^	0.76 ^†^	2.05	1.83 ^#^	1.49
Control Biscuit	2.7 ± 0.1	0.37	2.5 ± 0.1	2.8 ± 0.1	2.4 ± 0.1	2.7 ± 0.1	2.8 ± 0.1	0.01 ^†^	−0.12	0.80 ^†^	-
Protein Biscuit	2.6 ± 0.1		2.5 ± 0.2	2.7 ± 0.1	2.4 ± 0.1	2.4 ± 0.1	2.8 ± 0.1	0.50 ^†^	1.43	2. 12 ^#^	0.85
**JAR Colour**											
Control Cake	3.0 ± 0.04	**<0.001**	3.1 ± 0.06 ^aA^	2.9 ± 0.05 ^b^	3.0 ± 0.07	3.0 ± 0.06 ^A^	3.0 ± 0.07 ^A^	0.91	0.16	−0.26	0.90
Protein Cake	2.8 ± 0.04		2.8 ± 0.06 ^B^	2.8 ± 0.05	2.9 ± 0.07 ^a^	2.7 ± 0.06 ^bB^	2.8 ± 0.07 ^aB^	0.57 ^†^	-	1.17	1.26
Control Biscuit	2.4 ± 0.1	**<0.0001**	2.2 ± 0.1 ^aA^	2.8 ± 0.1 ^bA^	2.5 ± 0.1 ^A^	2.4 ± 0.1 ^A^	2.4 ± 0.1 ^A^	−0.61	-	0.30 ^†^	0.40
Protein Biscuit	3.2 ± 0.1		3.3 ± 0.1 ^B^	3.3 ± 0.1 ^B^	3.3 ± 0.1 ^B^	3.2 ± 0.1 ^B^	3.2 ± 0.1 ^B^	1.46	1.00	0.27	0.18 ^†^

Values are expressed as LSM estimates ± standard error from SAS output. Significant differences (*p* < 0.05) within a row (i.e., age YA vs. OA and saliva flow pairwise comparisons) are denoted by differing small letters, and within a column (i.e., within an age group between samples or within saliva flow groupings between samples) are denoted by differing capital letters. In penalty analysis # represents a significant difference (*p* < 0.05) between penalty analysis groups within a sample for older adults, † represents where the size of the group is lower than 20% of the population, and - represents where 0% of the population selected the category. Individual saliva flow groupings are derived from unstimulated saliva flow only, through tertile analysis.

**Table 8 foods-09-01328-t008:** Examples of volunteer comments in both studies.

Sample	Comments and Volunteers Details
Control Cake ^1^	*Pretty soft and with a nice edge easy to chew and swallow. Not as grainy as the last sample (v38, male, younger adult, aged 25). A nice moist cake with a good flavour. Very easy to eat (v58, male, older adult, aged 77).*
Protein Cake ^1^	*It is missing fluffiness, it feels a lot like a sponge with too many air bubbles (v12, female, younger adult, aged 24). Dry, rubbery, tasteless (v63, female, older adult, aged 76).*
Control Biscuit ^1^	*Really nice, I would buy this (v31, female, younger adult, aged 29). A very nice tasty biscuit, excellent and a very good flavour (v66, male, older adult, aged 81).*
Protein Biscuit ^1^	*Disliked the flavour, too artificial, unpleasant aftertaste (v33, female, younger adult, aged 21). The combination of lack of distinctive flavour and texture is not appealing and makes it unattractive (v86, female, older adult, aged 68).*
Control Cupcake ^2^	*Flavour is lovely and tasty (v31, female, younger adult, aged 28). Good texture and pleasant mouthfeel. Moist. Enjoyable to chew. I was surprised as I didn’t particularly like the appearance of the sample when I first opened it (v62, female, older adult, aged 73)*
Protein Cupcake ^2^	*Lovely flavour, just missing icing (v22, younger adults, aged 21). Very dry. Pleasantly chewy. Quite sweet but without much flavour (v73, male, older adult, aged 78).*

^1^ Pilot study; ^2^ main study.

**Table 9 foods-09-01328-t009:** Volunteers’ appetite mean ratings (change from baseline) of cupcakes in the main study; overall and by age and unstimulated saliva flow rate.

	Overall (*n* = 70)		Age		Unstimulated Saliva Flow		
		Significance of Sample (*p*-Value)	Younger Adults (*n* = 38)	Older Adults (*n* = 32)	Low Saliva Flow (*n* = 23)	Medium Saliva Flow (*n* = 23)	High Saliva Flow (*n* = 24)
**Hungry**							
Control Cupcake	−10.8 ± 3.7	1.00	−14.4 ± 5.3	−7.1 ± 4.5	13.0 ± 5.5	7.1 ± 5.2	12.2 ± 5.7
Protein Cupcake	−13.1 ± 3.7		−9.5 ± 5.2	−16.8 ± 4.5	10.4 ± 5.4	15.0 ± 5.2	14.0 ± 5.7
**Thirsty**							
Control Cupcake	9.4 ± 4.0	**0.04**	15.1 ± 5.8	3.6 ± 4.8	6.3 ± 6.0 ^aA^	15.1 ± 5.6 ^b^	6.6 ± 6.1 ^a^
Protein Cupcake	15.5 ± 4.0		16.1 ± 5.7	15.0 ± 4.8	18.2 ± 5.9 ^B^	11.9 ± 5.5	16.4 ± 6.1
**Desire to Eat**							
Control Cupcake	−14.9 ± 3.7	0.10	−20.8 ± 5.2 ^aA^	−9.0 ± 4.4	12.0 ± 5.5	9.5 ± 5.1	23.2 ± 5.6
Protein Cupcake	−19.0 ± 3.7		−20.7 ± 5.2 ^aA^	−17.4 ± 4.4	20.8 ± 5.3	18.2 ± 5.1	18.2 ± 5.6
**Satiety**							
Control Cupcake	6.7 ± 4.1	0.27	7.5 ± 5.9 ^aA^	6.0 ± 4.9	1.7 ± 6.1	4.0 ± 5.7	14.6 ± 6.2
Protein Cupcake	7.5 ± 4.0		3.6 ± 5.8 ^aA^	11.4 ± 4.9	10.2 ± 6.0 ^a^	3.5 ± 5.6 ^b^	8.8 ± 6.2 ^a^
**Fullness**							
Control Cupcake	9.8 ± 3.9	0.48	13.9 ± 5.6	5.7 ± 4.7	5.0 ± 5.9 ^A^	12.5 ± 5.6	12.0 ± 6.0
Protein Cupcake	8.0 ± 3.9		6.2 ± 5.5	9.8 ± 4.7	9.6 ± 5.8 ^aB^	1.4 ± 5.5 ^b^	13.1 ± 6.0 ^a^
**Prospective Consumption**							
Control Cupcake	−4.2 ± 3.3	0.45	−4.9 ± 4.8	−3.6 ± 4.0	−2.2 ± 5.0 ^a^	7.0 ± 4.7 ^b^	7.9 ± 5.0 ^a^
Protein Cupcake	−5.8 ± 3.3		−0.4 ± 4.7 ^a^	−11.0 ± 4.0 ^b^	−3.7 ± 4.9	9.2 ± 4.7	11.7 ± 5.1

Values are expressed as LSM estimates ± standard error from SAS output. Significant differences (*p* < 0.05) within a row (i.e., age YA vs. OA and saliva flow pairwise comparisons) are denoted by differing small letters, and within a column (i.e., within an age group between samples or within saliva flow groupings between samples) are denoted by differing capital letters. Appetite ratings were measured on a VAS (0–100 mm) and reflect a change from baseline (positive/negative values relate to the specific appetite rating being measured, for example, a negative hunger rating represents a decline in hunger). Individual saliva flow groupings are derived from unstimulated saliva flow only, through tertile analysis.

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
