# Peer review of "Consistent Effects of Whey Protein Fortification on Consumer Perception and Liking of Solid Food Matrices (Cakes and Biscuits) Regardless of Age and Saliva Flow"

_foods, 2020, doi:10.3390/foods9091328_

Round 1
Reviewer 1 Report
The manuscript investigates the effect of protein fortification of cakes and biscuits in order to study the relationship between the individual differences in terms of salivary flow rates, dental status and mouth behavior of two groups of consumers (young vs older adults) and the acceptance and perception of the fortified foods.
The studied topic is worth of interest and opens to new research regarding the sensory perception of food and food oral processing. The experimental plan is appropriate and the results are very well presented and discussed. Even if the hypothesis tested by the authors seems not confirmed, the study provides new scientific knowledge interesting for future research.
Specific comments:
L160. Please, specify the water temperature.
L243. Please provide a suitable reference for tertiary analysis.
Table 4. Values are expressed as mean ± standard deviation while in Tables 6-7 and Figures 3-5 values are reported as mean ± standard error. I suggest uniforming the data presentation, expressing the values as mean ± standard error also in Table 4.
L482. It is reported that no instrumental difference in colour were instrumentally found for cakes while both the trained panel and the volunteers perceived differences in colour between the control and protein cakes. Based on my experience, it is common to detect instrumental colour differences that are not detected by human subjects but not the opposite. Please, comment and try to explain your result. I would suggest to consider adding as colour measure also the “colour difference (DE) and check if the obtained results will be lower than the just noticeable difference (JND; DE =2.2).
Author Response
Thank you for your comments relating to our paper ‘Consistent effects of whey protein fortification on consumer perception and liking of solid food matrices (cakes and biscuits) regardless of age and saliva flow’. We have addressed all of your and tracked all changes in the manuscript. Below is a summary of the changes made:
L160. Please, specify the water temperature.
Response: Thank you for your comment. We have added water temperature (warm filtered water ~ 40 °C) to Line 197.
L249. Please provide a suitable reference for tertiary analysis.
Response: We have reworded this as Tertiles rather than tertiary to be less confusing. “Volunteers were categorised into low, medium and high groups based on average unstimulated salivary flow rates, using tertiles in XLSTAT; these groupings were also used for subsequent statistical analysis”.
Table 4. Values are expressed as mean ± standard deviation while in Tables 6-7 and Figures 3-5 values are reported as mean ± standard error. I suggest uniforming the data presentation, expressing the values as mean ± standard error also in Table 4.
Response: Thank you for your comment. We have updated Table 4 to standard error.
L482. It is reported that no instrumental difference in colour were instrumentally found for cakes while both the trained panel and the volunteers perceived differences in colour between the control and protein cakes. Based on my experience, it is common to detect instrumental colour differences that are not detected by human subjects but not the opposite. Please, comment and try to explain your result. I would suggest to consider adding as colour measure also the “colour difference (DE) and check if the obtained results will be lower than the just noticeable difference (JND; DE =2.2).
Response: Thank you for this, very helpful. We have calculated the colour difference and added to Table 4. The colour difference are above the JND however this does not explain the substantial perceptual differences in the cakes. We propose the instrumental colour measurements are underestimating the perceptual difference between cakes due to the aeration of the sample and have added this to the results (text above table 4), as well as slightly modifying the discussion.

Reviewer 2 Report
Can authors please indicate numbers of assessors used in hedonic tests in the abstract. This need to be clearly outlined from the start. Also, chemo-sensory losses are a major and fundamental issue for elderly consumers. Authors should at least mention in the introduction that this is considered, even better any work previous studies undertaken on cakes. Michon, C (FQP) did some work in 2010 on this.
Author Response
Thank you for your comments relating to our paper ‘Consistent effects of whey protein fortification on consumer perception and liking of solid food matrices (cakes and biscuits) regardless of age and saliva flow’. We have addressed your comments and tracked all changes in the manuscript. Below is a summary of the changes made:
Can authors please indicate numbers of assessors used in hedonic tests in the abstract. This need to be clearly outlined from the start. Also, chemo-sensory losses are a major and fundamental issue for elderly consumers. Authors should at least mention in the introduction that this is considered, even better any work previous studies undertaken on cakes. Michon, C (FQP) did some work in 2010 on this.
Response: Thank you for your comments.
We have added in the abstract the number of assessors as follows:
The sensory profile and physical properties were analysed and two volunteer studies (n = 84; n = 70) were carried out using two age groups (18-30; 65+) (Line 20/21).
We have added chemosensory losses to the introduction and added the Michon et al. (2010) reference as follows:
Food intake can also be reduced in older adults due to chemosensory impairments (such as loss of taste and smell) which influence food choices and consumption [6] (Line 41/42).
In addition, previous research has suggested older adults had higher liking for cakes compared with younger adults [12] therefore supporting the use of popular products (such as cakes) to potentially increased food intake within older adults (Line 54-56).

Reviewer 3 Report
The manuscript aims to explore how the protein fortified products is associate with negative sensory attributes and poor consumer acceptance. The application of solid model to cakes and biscuits were investigated, but the analysis of sensory profile and physical properties should be better explained.
The manuscript is not well structured. It needs more clear structure on introduction, review of literature, materials and methods, empirical results and discussion, implications, and conclusion. In addition, the manuscript needs some clarifications and improvements:
Abstract:
In the abstract there should more clearly stated the main aims, possible novelties and/or contributions, main findings, and implications.
1. Introduction
The section on the Introduction is without clear motivation. It is suggested to specify in a better way the hypothesis and possible novelty and/or contribution of the manuscript to the literature. Introduction should be brief, providing motivation of the research and outline main research focus. The objectives must be specified more in detail. Second, in the literature review, there is missing a strong section on review of recent relevant literature. There is emerging literature on the subject.
For example, at lines 43-44 is reported how the protein requirements are considered to increase with age and are associated with many positive functional outcomes. Why? Please cite the literature.
2. Materials and Methods
The explanations relating to the materials used for the analyses are dispersive and a little confusing. I suggest summarizing the subparagraphs 2.3, 2.4 and 2.7. Despite this, I suggest to improve the paragraph 2.8. For example, for the application of the ANOVA see the following work:
Fanelli R.M., Di Nocera A. (2018). Customer perceptions of Japanese foods in Italy. JOURNAL OF ETHNIC FOODS, 5, 167-176. https://doi.org/10.1016/j.jef.2018.07.001
3. Results
The results should be summarized, illustrated with more emphasis and should be deepened in relation to dental status and age in both studies. Please explain in a better way why the age had no significant effect on stimulated saliva flow (line 320), Figure 4.
What are the study implications?
How qualitative feedback can help the producers of cakes and biscuits to make these two products more palatable
4. Discussion
The discussion can improve with the following clarifications:
How in the previous literature it has been shown that the control and protein cupcake had the same moisture content? (lines 471-472).
How findings can supportive of developing foods for the needs of older adults in mind? (line 523)
Please, explain how to better understand the causes of poor compliance and consumer acceptance and how this is very important for operators in the confectionery sector (lines 560-561)
5. Conclusions
Conclusions should be improved as they largely repeated the results. The character of conclusion is too general. Authors should better underline conclusions, and intensions for future researches
What are the study limitations?
What are the proposals for research in future?
Finally, Regarding Tables and Figures: the quality should be improved.
Author Response
Thank you for your comments relating to our paper ‘Consistent effects of whey protein fortification on consumer perception and liking of solid food matrices (cakes and biscuits) regardless of age and saliva flow’. We have addressed your comments and tracked all changes in the manuscript. Below is a summary of the changes made:
Abstract:
In the abstract there should more clearly stated the main aims, possible novelties and/or contributions, main findings, and implications.
Response: Thank you for your comment, we have reworded the abstract.
- Introduction
The section on the Introduction is without clear motivation. It is suggested to specify in a better way the hypothesis and possible novelty and/or contribution of the manuscript to the literature. Introduction should be brief, providing motivation of the research and outline main research focus. The objectives must be specified more in detail.
Second, in the literature review, there is missing a strong section on review of recent relevant literature. There is emerging literature on the subject. For example, at lines 43-44 is reported how the protein requirements are considered to increase with age and are associated with many positive functional outcomes. Why? Please cite the literature.
Response: Thank you for your comments. Concerning the guidelines to increase protein intake in older age we have added the predominant functional outcomes and referred to the Prot-age study. We have added a sentence concerning chemosensory changes with age, and a study referring to cakes and older adults to justify why cakes and familiar liked products are sensible to fortify. The key aims of the study were to confirm whether protein fortification of cakes and biscuits increases mouthdrying, and to evaluate whether this is effected by age and individual factors including saliva flow rate.
- Materials and Methods
The explanations relating to the materials used for the analyses are dispersive and a little confusing. I suggest summarizing the subparagraphs 2.3, 2.4 and 2.7. Despite this, I suggest to. improve the paragraph 2.8. For example, for the application of the ANOVA see the following work:
Fanelli R.M., Di Nocera A. (2018). Customer perceptions of Japanese foods in Italy. JOURNAL OF ETHNIC FOODS, 5, 167-176. https://doi.org/10.1016/j.jef.2018.07.001
Response: We agree that the methods section concerning cake and biscuit preparation (section 2.3) was too wordy and we have reduced this. We believe that combining the subsections into less section would be more confusing for the reader so we have not reduced the number of sections. We could find no further details that could be removed in sections 2.4 to 2.7.
Thank you for suggesting the Fanelli paper which we have read. However, regarding paragraph 2.8 and the application of ANOVA, we have used ANOVA for QDA data as this is considered the most appropriate method (as supported by references we give in the paper). We have used software specifically designed for sensory profiling data (Senpaq) and used the recommended method for sensory panels which is a mixed model where samples are treated as fixed effects and panellists and random effects. For the physical properties data we were comparing only two samples with one treatment effect (sample) therefore t-tests were used rather than ANOVA where residuals were normally distributed, and the Mann-Whitney test where they were not. For the data from volunteers we used a linear mixed model which we explain comprehensively in section of 2.8.
- Results
The results should be summarized, illustrated with more emphasis and should be deepened in relation to dental status and age in both studies. Please explain in a better way why the age had no significant effect on stimulated saliva flow (line 320), Figure 4.
What are the study implications?
How qualitative feedback can help the producers of cakes and biscuits to make these two products more palatable
Response: Thank you for your feedback. In the results section we have summarised all results relating to both products (cakes and biscuits) and individual differences (such as age, dental status, etc.), we did try to avoid over-discussing the results in this section and avoid duplication of the later discussion. Age did effect drying perception of the products as stated in the results section (lines 407 onwards) and later discussed in section 4.2. Dental status did significantly effect liking and easiness to eat and swallow scores as stated in lines 426-429, and discussed in section 4.2. However, other individual differences had only minimal effects on perception and liking therefore we felt it was not appropriate to overstate results. Please note further data on dental status is outlined in supplementary materials.
We have added a rationale in the discussion about no effect of stimulated saliva flow on age as follows:
“However, our study demonstrated no age-related differences in stimulated salivary flow rates, supporting our previous work [19]. Furthermore, it is proposed that age-related changes in salivary flow rates is gland specific with the parotid and minor salivary glands potentially being less influenced by age [49].” (Lines 597-600)
- Discussion
The discussion can improve with the following clarifications:
How in the previous literature it has been shown that the control and protein cupcake had the same moisture content? (lines 471-472).
Response: Following the pilot study we optimised the cakes to match the moisture content and this was not based on previous literature but on adaption of the formulation (Line 528).
How findings can supportive of developing foods for the needs of older adults in mind? (line 523)
Response: We have reworded the sentence to the following:
Accordingly, these findings are supportive of developing foods appropriate for older adults, for example, those with reduced dental status (lines 583-584).
Please, explain how to better understand the causes of poor compliance and consumer acceptance and how this is very important for operators in the confectionery sector (lines 560-561).
Response: We have reworded the sentence to the following:
“Given the derived benefits from ONS and protein fortified products, it is important to understand the causes of poor compliance and consumer acceptance, which have been associated with mouthdrying and off flavours (lines 624-626)”.
- Conclusions
Conclusions should be improved as they largely repeated the results. The character of conclusion is too general. Authors should better underline conclusions, and intensions for future researches
Response: We have reworded to improve clarification and more emphasis on future research.
What are the study limitations?
Response: We have added a limitation section (section 4.3).
What are the proposals for research in future?
Response:
This is summarised in the final two sentences of the conclusion in addition to highlighting other future research in the discussion (last paragraph of section 4.1 and 4.2).
Finally, Regarding Tables and Figures: the quality should be improved.
Response: Figure 3 unfortunately moved once uploaded to the foods journal system, we have now addressed this.

Round 2
Reviewer 3 Report
None